# Multishell Diffusion MR Tractography Yields Morphological and Microstructural Information of the Anterior Optic Pathway: A Proof-of-Concept Study in Patients with Leber’s Hereditary Optic Neuropathy

**DOI:** 10.3390/ijerph19116914

**Published:** 2022-06-05

**Authors:** David Neil Manners, Laura Ludovica Gramegna, Chiara La Morgia, Giovanni Sighinolfi, Cristiana Fiscone, Michele Carbonelli, Martina Romagnoli, Valerio Carelli, Caterina Tonon, Raffaele Lodi

**Affiliations:** 1Department of Biomedical and Neuromotor Sciences (DIBINEM), University of Bologna, 40138 Bologna, Italy; lauraludovica.gramegna@unibo.it (L.L.G.); giovanni.sighinolfi3@unibo.it (G.S.); cristiana.fiscone2@unibo.it (C.F.); michele.carbonelli@ausl.bologna.it (M.C.); valerio.carelli@unibo.it (V.C.); caterina.tonon@unibo.it (C.T.); raffaele.lodi@unibo.it (R.L.); 2IRCCS Istituto delle Scienze Neurologiche di Bologna, Bellaria Hospital, 40139 Bologna, Italy; chiara.lamorgia@unibo.it (C.L.M.); martina.romagnoli@isnb.it (M.R.)

**Keywords:** Leber’s hereditary optic neuropathy, anterior optic pathway, brain tractography, brain functional morphology, brain diffusion-weighted MR imaging, neuro-ophthalmology, human anatomy of the nervous system

## Abstract

Tractography based on multishell diffusion-weighted magnetic resonance imaging (DWI) can be used to estimate the course of myelinated white matter tracts and nerves, yielding valuable information regarding normal anatomy and variability. DWI is sensitive to the local tissue microstructure, so tractography can be used to estimate tissue properties within nerve tracts at a resolution of millimeters. This study aimed to test the applicability of the method using a disease with a well-established pattern of myelinated nerve involvement. Eight patients with LHON and 13 age-matched healthy controls underwent tractography of the anterior optic pathway. Diffusion parameters were compared between groups, and for the patient group correlated with clinical/ophthalmological parameters. Tractography established the course of the anterior optic pathway in both patients and controls. Localized changes in fractional anisotropy were observed, and related to estimates of different tissue compartments within the nerve and tract. The proportion of different compartments correlated with markers of disease severity. The method described allows both anatomical localization and tissue characterization in vivo, permitting both visualization of variation at the individual level and statistical inference at the group level. It provides a valuable adjunct to ex vivo anatomical and histological study of normal variation and disease processes.

## 1. Introduction

Study of cranial nerve anatomy has historically been performed by dissection of donated specimens. Anatomy textbooks provide a valuable tool for learning as they provide a schematic and stereotypic representation of each tract. However, their ability to represent three-dimensional anatomy is limited as nerves pass through multiple slices in any given plane. Finally, interactive tools based on computer representations of anatomy can bridge the gap between these two methods. Tomographic scans allow both practicing clinicians and students to visualise the anatomy of individual subjects, not only of representative individuals but patients whose anatomy may vary from that presented in textbooks. Magnetic resonance (MR) imaging above all excels in soft tissue visualisation [1], although the cranial nerves present a particular challenge due to their morphology (elongated in one dimension and restricted in diameter), and the vicinity of other tissue types that present as iso-intense on widely used MR sequences or modify the apparent intensity of surrounding tissue [2].

MR tractography can mitigate this problem as it allows a three-dimensional visualization of the cranial nerve pathway while simultaneously evaluating the microstructural properties of the tissue therein.

As a proof of principle, a group of participants including both healthy adults and patients with optic neuropathy was imaged, to test the utility of performing magnetic resonance tractography of the whole anterior-optic pathway, including the optic nerve, to obtain anatomical and microstructural information. In particular, we evaluated patients affected by Leber’s hereditary optic neuropathy (LHON).

Briefly, the anterior optic pathway comprises the retinal ganglion cells (RGCs), the optic nerve itself, the median merging of the optic nerves into the optic chiasm, and the optic tracts that reach the lateral geniculate nucleus. In normal anatomy, RGC axons from the retina converge and come together at the optic nerve head, forming what at fundus exam appears as the optic disc. In the so-called retinal nerve fibre layer (RNFL) the RGC axons are unmyelinated, their conduction velocity is therefore much slower than for typical axons, and they require far greater energy to restore electrical potential following discharge, being among the highest oxygen consuming tissues of the body with high numbers of mitochondria. After leaving the optic head and having crossed the lamina cribrosa, the optic nerve fibres then comprise both axon and a myelin sheath, the latter being composed mainly of oligodentrocytes. This skewed anatomical structure of the optic nerve axons with the unmyelinated high-energy demand section as opposed to the myelinated portion of the optic nerve makes the RGCs particularly vulnerable to mitochondrial dysfunction. About 5 cm posterior to the eye, as mentioned, the two optic nerves merge to form the optic chiasm, where there is a partial decussation, involving just over half the fibres. Only the axons from the nasal retina (temporal visual field) cross over joining the contralateral optic nerve’s temporal retinal fibres and form the contralateral optic tract. From the optic tract, most fibres terminate in one of six layers of the lateral geniculate nucleus.

Leber’s hereditary optic neuropathy (LHON) is one of the most common inherited optic neuropathies with an estimated prevalence of about 1:25,000–60,000 inhabitants [3,4,5]. It is associated in over 90% of cases with one of three mtDNA missense point mutations—m.3460G > A, m.11778G > A, and m.14484T > C—in genes encoding for the mitochondrial respiratory chain complex I subunits [6]. The disease is characterized by a sudden, subacute visual loss due to selective loss of RGCs and relative sparing of the melanopsin-expressing RGCs, leading to variable degree of optic nerve atrophy and different degrees of blindness in young adulthood [3,7]. Conventional MRI is normal in the majority of patients with LHON, even though optic nerve and visual pathway hyperintensity have also been reported; the pathophysiology of the MRI changes is not yet understood [8,9].

In postmortem LHON cases, a complete loss of temporal fibres, serving central vision, with various degrees of axonal sparing in the periphery at the level of the optic nerve cross-section has consistently been found [10,11]. Moreover, in the myelinated portion of the optic nerve a wide variability of myelin thickness is also evident, with some axons being almost denuded of myelin sheath. In some cases, evidence of remyelination has been recognized, suggesting a dynamic physiopathological process rather than simply neurodegeneration [3,12,13].

Diffusion-weighted imaging is an MR modality that allows the identification and visualisation of tissue types that present microstructural anisotropy, such as myelinated neurons. Probabilistic tractography extends this analysis to create maps that are derived from the probability that two regions are connected, in this case the terminal portion of the optic nerve and the lateral geniculate nucleus (LGN) [14]. No previous study has applied tractography to the evaluation of LHON patients over the whole anterior optic pathway including the optic nerve [15,16].

The aim of the current study was to test whether tractography in the optic tract is feasible for the aim of visualising its course in three dimensions and simultaneously extracting microstructural information useful for demonstrating pathological variation. We hypothesise that it is possible to delineate the optic tract between the lateral geniculate nucleus and the orbit in both healthy subjects and LHON patients, and that an along-tract analysis will demonstrate differences at the group level that correlate with pathology.

## 2. Materials and Methods

This is a prospective, observational MRI study of well-characterized LHON patients along with age-matched controls. 

### 2.1. Participants

All patients with a molecular diagnosis of Leber’s hereditary optic neuropathy and no additional atypical findings referred to the Neuroimaging Laboratory by the Laboratory of Neurogenetics at the Institute of Neurological Sciences of Bologna, Bellaria Hospital, Bologna, Italy, between April 2021 and January 2022, were considered to be eligible for evaluation. Written informed consent was obtained from patients prior to participation in the study, which was conducted according to the guidelines of the Declaration of Helsinki and approved by the “Area Vasta Emilia Centro” Ethics Committee (CE-AVEC) with approval number NOE-121/2019/OSS/AUSLBO—CE 19012. Participants underwent a clinical and neuro-ophthalmological assessment and a standard MRI scan protocol. A group of healthy adults with a matched age range were recruited from the database of the Neuroimaging Laboratory (IRCCS Institute of Neurological Sciences of Bologna), designed to collect normative values of quantitative MR parameters for clinical and research purposes.

Neuro-ophthalmological assessment included the evaluation of the best corrected visual acuity (VA) by ETDRS charts and optical coherence tomography (OCT) for assessing the average retinal nerve fibre layer (RNFL) thickness (OCT: DRI Triton, Topcon, Tokyo, Japan). We also retrieved information regarding disease duration for all patients included in the study. Neuro-ophthalmological and MRI examinations were conducted not more than 3 days apart.

### 2.2. Magnetic Resonance Imaging

The MRI protocol was performed using a Siemens MAGNETOM Skyra 3-T MRI scanner equipped with a 64-channel head–neck array coil.

The MRI protocol included volumetric T1-weighted imaging based on 3D MPRAGE; 176 continuous sagittal slices were acquired, with 1 mm isotropic voxel, no slice gap, echo time (TE) = 2.98 ms, repetition time (TR) = 2300 ms, inversion time (IT) = 900 ms, flip angle (α) = 9°, acquisition matrix = 256 × 256, pixel bandwidth = 240 Hz, in-plane acceleration factor = 2, scan duration approximately 5 min.

For tractography analyses, a HARDI diffusion-weighted protocol was acquired based on a 2D single-shot echo planar imaging (EPI) sequence; 87 continuous axial oblique slices were acquired aligned with the bicommisural axis, with 2 mm isotropic voxel, no slice gap, TE/TR = 98 ms/4300 ms, α = 90°, acquisition matrix = 110 × 110, pixel bandwidth = 1820 Hz, in-plane acceleration factor = 2, multiband acceleration factor = 3, phase encoding anterior–posterior (AP), scan duration approximately 9 min.

The diffusion-weighting schedule was designed based on [17]. Briefly, the following EPI volumes were acquired: b = 0 (n = 8), b = 300 s mm^−2^ (n = 8), b = 1000 s mm^−2^ (n = 30), and b = 2000 s mm^−2^ (n = 64), all with an anterior–posterior (AP) phase-encoding direction, repeating each volume acquisition with weighting 0–1000 s mm^−2^ with inverted (PA) phase-encoding direction, scan duration approximately 4 min. The information from the two phase-encoding directions was used to correct EPI distortion artefacts in the EPI volumes.

### 2.3. Morphological Assessment of the Chiasm

In order to determine whether conventional imaging could contribute to the assessment of morphological assessment of the optic tract, one neuroradiologist with over 10 years of practice (L.L.G.) measured the diameters of the intracranial optic nerve, of the optic chiasm and of the optic tracts of the axial reformatted image, following a previously validated method [18].

### 2.4. Imaging Preprocessing

Data from each participant were preprocessed using an in-house automated pipeline developed from software packages freely available as part of the Oxford Functional Magnetic Resonance Imaging Software Library (FSL) [19], version 6, the NIH Functional Neuroimaging library (AFNI) [20], version 20.3.02, and MRtrix, version 3.0.2 [21].

Diffusion-weighted images were skull-stripped using the FSL bet tool. Image denoising was performed with the MRtrix3 dwidenoise function [14], using a principal component analysis approach. Susceptibility-related distortions in the EPI acquisition were estimated using the FSL topup function; subsequently, a combined correction for susceptibility, eddy-current effects, and signal dropout, most commonly induced by subject movement, was performed by FSL eddy_openmp, based on the topup estimates.

The DWI data were modelled at the voxel level using functions of the MRtrix toolkit. In particular, the response function for spherical deconvolution was modelled using the Dhollander algorithm within dwi2fod [22,23]. Fibre orientations distributions were estimated using multishell multitissue spherical deconvolution (msmt_csd) to infer three parameters from three tissue compartments, two with isotropic diffusion characteristics, one representing restricted diffusion and labelled ”grey matter” (GM) and the other representing unhindered diffusion and labelled cerebrospinal fluid (CSF); finally one component with anisotropic characteristics modelled as a tensor, was labelled ”white matter” (WM) [24]. A fourth component was calculated subsequently to account for voxels in which the total absolute intensity did not reach 0.075; it represented tissues such as bone that did not generate MR signal, or in which no tissue was present. This component was labelled ”background”. The intensity of each component was normalised so that in total the four components summed to unity. The normalised components were labelled nGM, nCSF, and nWM. An example of the distribution of the four compartment fractions is shown in Appendix A.

Further characterization of the diffusion properties of tissue within each voxel was provided by FSL dtifit based on the diffusion tensor, which was used to estimate voxel fractional anisotropy (FA) and mean diffusivity (MD).

### 2.5. Tract Estimation

Tractography is a method that converts volumetric maps of diffusion characteristics into estimates of the probable location white matter tracts, based on the generation of streamlines linking adjacent voxels. The MRtrix tool tckgen generated streamlines using a first-order integration over fibre orientation distributions (iFOD1) approach, following [25].

In order to isolate the optic pathway, selected fibres were projected back to the DWI imaging space to create a volumetric probability map. Voxels were thresholded to exclude low probability tracts as follows. For each coronal slice containing at least one fibre, the maximum number of fibres in any voxel was identified, and all voxels containing less than 40% of the maximum were excluded. Because the tractography algorithm attempts to place tracts in low probability regions where the anisotropic component is not evident, an exclusion mask was created by identifying regions where normalised background was present. The initial threshold was 15% and reduced iteratively until a continuous tract was defined that reached the waypoint in the postorbital tract.

Tractography was deemed to be successful if a continuous tract was generated, if there was substantial overlap between the paths of decussating and nondecussating fibres, and if the proportion of nondecussating fibres was close to the value attested in the literature (around 50%).

### 2.6. Statistical Analyses

For the purposes of visualisation, the fibre density maps for each individual were superimposed onto their own registered T1-weighted image, using the MRtrix program MRview [21]. Each map was also projected into MNI space to generate an average distribution map of the whole sample.

As the principal direction in the pregeniculate portion of the optic tract is anterior–posterior, along-tract statistics were calculated over successive coronal slices. For each slice, the mean value within the tract of fractional anisotropy (FA), mean diffusion (MD), and normalised volume fraction for each of the three compartments with isotropic restricted and unrestricted diffusion and anisotropic diffusion (nCSF, nWM, nGM, as defined in Section 2.4) was calculated, along with the number of voxels containing at least one fibre, and the maximum fibre number normalised by the number of seed voxels. As parameter estimation sometimes fails at the voxel level, values outside a fixed range were assumed to be inaccurate and not included in the calculation of mean values, as follows: FA outside (0, 1), MD outside (0, 0.005).

To compare parameters between individuals, data were binned by grouping slices such that the portion of the tract between the LGN and the chiasm was divided into ten evenly spaced segments, and the portion from the chiasm to the postorbital seed into seven segments, with lengths adjusted to the cranial dimensions of each subject. Average segment length was around 4 mm (Figure 1b,d). Within each segment, patient and healthy control groups were compared using Welch’s t-test. Results with *p*-values below 0.05 after familywise error correction by the Hochberg step-up procedure were considered to be significant.

In an exploratory analysis, patients’ clinical/ophthalmological data were compared to diffusion parameters that demonstrated differences between LHON and HC groups. Within the optic tract, MRI data in an ROI comprising the whole left or right tract were compared to clinical data from the ipsilateral eye. In regions defined within or posterior to the chiasm, where fibres from nerves of both sides are present, each tract was compared to clinical data grouped as first affected or second affected eye. In each case, Pearson’s correlation coefficient was calculated. Results are reported when the uncorrected *p*-value fell below a pragmatic threshold of 0.05.

## 3. Results

Eight LHON patients prospectively enrolled in the study between April 2021 and January 2022 were referred to the neuroimaging laboratory. All carried the homoplasmic m.11778G > A/MT-ND4 mitochondrial DNA mutation. 

Seven of eight patients were male. The mean age of patients was 32.8 years (standard deviation, s.d. 16.2 years). In the LHON group, the mean interval between the onset of symptoms and the MR acquisition (the disease duration) was 332 (s.d. 315) days for the right eye and 315 (s.d. 126) days for the left eye. Visual acuity on the logMAR scale was 1.69 (s.d. 0.34) for the right eye, and 1.69 (s.d. 0.36) for the left. Average RNFL thickness was 76.2 (s.d. 22.0) μm for the right eye and 74.9 (s.d. 22.2) μm for the left. For the postchiasmatic tract the average RNFL thickness of the two contributing lateral quadrants was used: right temporal and left nasal for the right side, left temporal and right nasal for the left side. 

The mean age of the control group it was 32.6 years (s.d. 11.2 years). Six out of thirteen controls were male.

### 3.1. Morphometric Analysis

Morphometric assessment of the optic nerve in the vicinity of the chiasm demonstrated no differences between patients and healthy subjects. The results are summarised in Appendix A. 

### 3.2. Tractography

Tractographic analysis was successful in 12/13 healthy subjects and 8/10 patients in all four tracts (crossing and noncrossing fibres originating from both left and right LGN), in the sense that streamlines could be tracked between the preselected starting and endpoints without interruption. In two patients and one healthy control, the noncrossing fibres on the left side did not arrive at the orbital waypoint. In these cases, tractography was successfully performed separately on the pre- and postchiasmatic optical fibre bundles, and the union of the two segments was used to define the entire tract.

The proportion of nondecussating fibres measured by the weighted fibre volume was similar between left and right tracts. For LHON patients, the mean fraction was 54.4% (s.d. 5.6%) on the left side and 50.1% (s.d. 6.3%) on the right, while for healthy controls the fractions were 49.6% (4.3%) on the left and 48.6% (3.3%) on the right. Comparing the degree of overlap between crossing and noncrossing tracts, the mean Dice coefficient was 0.817 (s.d. 0.049) in healthy controls, and 0.658 (s.d. 0.219) in LHON patients. The lower degree of overlap in the patients was due mainly to the poor congruence in the two subjects in which tractography was performed piecewise.

Visualisation of individual tracts was performed using MRView, allowing for image reslicing in any desired plane. The majority of the tract remains with a doubly oblique sagittal slice (Figure 1), so that after rotation of the image volumes the course of the anterior pathway is easy to follow using either a slicewise or a three-dimensional projection. 

### 3.3. ROI Statistical Analysis

Differences in diffusion parameters between LHON and HC groups are summarised in Table 1. Fractional anisotropy of the LHON patient group is significantly reduced compared to controls in both the optic nerve and tract, and within the chiasm. Mean diffusivity is unremarkable in the patients compared to the control group. Turning to the compartment fraction estimates based on multishell diffusion analysis, the grey matterlike fraction is increased in patients both in the chiasm and bilaterally in postchiasmatic ROIs, while the white matter fraction is significantly decreased in 4/5 ROIs. 

### 3.4. Along-Tract Analysis

An along-tract analysis of diffusion parameters, dividing the tract into aligned 4 mm segments aids visualisation of the ROI results presented above (Figure 2). It shows that the MD (Figure 2a) exhibits considerable variation along the tract, but values in patients closely track those of the control group. On the other hand, FA (Figure 2c) is rather variable along the length of the tract, but in all segments other than those in the vicinity of the LGN, just anterior to the chiasm and the portion of the nerve closest to the orbit, values are significantly reduced in the patient group. 

Looking at the results of the multishell analysis, the variation in the nCSF fraction (Figure 2b) mirrors that of the mean diffusivity, and again there is almost no difference between the LHON and control groups. The normalised grey matterlike fraction (nGM, Figure 2d) is elevated in patients almost everywhere, except for a small segment about 4–20 mm anterior to the chiasm. The grey matterlike fraction is notably uniform in the control group in this segment and those relatively anterior, while in the patient group the fraction rises to a peak around 20 mm posterior to the most anterior segment, placed just behind the optic disc. In contrast the normalised white matterlike fraction (nWM, Figure 2e) is lower in patients along much of the length of the tract, notably in the segments within and posterior to the chiasm. Within the nerve the difference is most significant in the segments between 8 and 24 mm posterior to the segment behind the optic disc.

### 3.5. Correlation Analysis

Exploratory analysis in the patient group revealed that normalised GM-like fraction appeared to correlate with patient age for ROIs covering both the nerve and the optic tract itself (r = 0.517, *p* = 0.04; r = 0.681, *p* = 0.004, respectively), while normalised WM correlated with age only in the postchiasmatic ROI (*p* = −0.812, *p* < 0.001). In the healthy control group, a very similar relationship between normalised GM and age was found (*p* = 0.51, *p* = 0.008) in the nerve. There was no relationship with age in the optic tract itself.

In patients, among the ophthalmological variables assessed, apparent correlations were observed only in the optic nerve itself, both with the nWM: visual acuity was positively correlated (r = 0.569, *p* = 0.021) while average RNFL was negatively correlated (−0.576, *p* = 0.02). The full table of calculated correlations is shown in Appendix A.

## 4. Discussion

In this study, multishell diffusion MRI was used to implement tractography of the anterior optic pathway. The procedure outlined was able to define both the left and right tracts, including both decussating and nondecussating fibres in both healthy subjects and LHON patients, allowing visualisation of the course of the anterior pathway at the level of the single subject and as a probabilistic map in the MNI standard reference space. Tractography is an indirect method for tracing the course of myelinated axonal tracts. The high degree of overlap between decussating and nondecussating fibres indicated by the Dice measure gives some confidence in the accuracy of the results, as does the proportion of streamlines that cross the chiasm: it was around 50% in both patients and controls, as expected.

Teaching the gross anatomy of complex structures such as the anterior optic pathways to students is a challenging task [26] and some authors have suggested that clinical anatomy should be continuously reappraised in light of the findings of the newest research methods [27]. In particular, three-dimensional (3D) visualizations offered by DTI tractography can be used not only for clinical assessment but also for teaching complex anatomy, based on both standard atlas representations and individual variation (as shown in Figure 1). In line with this suggestion, a recent review supported the effectiveness of 3D visualization, yielding superior student test results compared to traditional methods [28].

The advantages of anatomical studies based on diffusion-weighted MRI are first that results can be obtained in individual subjects in vivo, and second that it is possible to extract microstructural information that can be referenced to specific anatomical locations. We initially hypothesised that such information could be used to uncover the presence of microstructural alterations within anterior optic pathways with apparently normal morphology. The nerve diameter in the vicinity of the chiasm, which has been reported as being sometimes enlarged in LHON [8], was no different in our study population. Diffusion parameters, evaluated either within macroscopic regions (nerve, chiasm, tract) or in short segments of average length 4 mm, did demonstrate interesting differences.

Considering first the more well-established diffusion tensor model of water diffusion within biological samples, fractional anisotropy, a parameter typically considered to indicate integrity of myelinated axons, was reduced in all macroscopic ROIs in the LHON group, and this finding was repeated in the along-tract analysis in all but a few segments, in particular those portions of the optic nerve where FA was low in both patients and controls. The mean diffusivity, on the other hand, was remarkably similar in the two groups, with some possible slight areas of increase in patients detectable in the along-tract analysis.

Multishell DWI methods such as high angular resolution diffusion imaging (HARDI) acquire a signal from tissue water with different degrees of diffusion weighting, and as such are sensitive to intra- and extracellular environments at different scales. This allows for multicompartment modelling of the signal from individual voxels. In this work, we used a model with three compartments, the first characterised by high isotropic diffusion, and conventionally labelled “CSF-like”, the second by restricted isotropic diffusion (“GM-like”) based on its distribution within brain images [29]. The third component, with anisotropic diffusion, is labelled, for the same reason, “WM-like”. We normalised the signal within the three compartments to a total of 100% with the inclusion of a nontissue compartment to allow the summation over voxels with very little signal, only background noise. Based on this parcellation, it is possible to obtain estimates of tissue contribution that have a more direct anatomical interpretation compared to the DTI parameters, FA and MD. It should be borne in mind, however, that the assignment is always based on diffusion characteristics. As stated by Mito [29], an increase in nGM fraction should not be thought of as an increase in actual grey matter, but a shift towards tissue with similar characteristics. Consequently, axonal tissue will be labelled as “GM-like” if it exhibits restricted diffusion with low anisotropy. Factors frequently cited for the appearance of anisotropic diffusion within nerves include myelination, axon density, and fibre coherence [30,31]. Decreased anisotropy could thus be due to a change in one or more of these factors. It is also compatible with astrogliosis within white matter as glial cells have diffusion characteristics more similar to grey matter [29].

Figure 2 makes clear the relationship between MD and the nCSF fraction. It shows that increases in MD are in large part explainable by the presence of tissues characterised by unrestricted water diffusion. The reduced nCSF fraction in the anterior part of the optic nerve behind the optic disc is probably due to partial volume with low-signal voxels, given that the fractions of tissue compartments (nGM and nWM) do not simultaneously increase. There are hints of an increase in nCSF in the optic tract that the current study does not have the power to resolve, but over all the data seem to suggest that there is no major change in axonal volumes in the LHON group.

The healthy optic nerve seems to be characterised by a low and stable nGM fraction (<0.2), while the nWM fraction increases to a peak about 12 mm behind the optic disc, and falls sharply thereafter, possibly due to decreased myelination or increased partial volume with nonaxonal tissue. In the LHON patients, the nGM fraction was elevated over most of the length of the nerve, apart from the section around 16 mm anterior to the chiasm, at the point where the nerve enters the intraconal space. The WM-like component was reduced over the central portion of the nerve, essentially in that part where the fraction in healthy subjects is elevated towards values seen in the optic tract (>0.35).

Posterior to the chiasm, the picture is simpler. As the tract follows it course posteriorly, in healthy controls the WM-like fraction increases to around 0.5 or above at the level of the mesencephalon. The GM-like component also increases, at the expense of the CSF compartment, though remaining below 0.4. In LHON patients, the proportions are reversed, and except in the segment closest to the LGN, the nWM fraction is less than the nGM fraction.

Correlation analyses suggest that in patients there is a decreasing fraction of WM-like tissue with age in the optic nerve and a parallel increase in nGM. There is also an age-related increase in the GM-like fraction along the length of the optic tract, but this does not reflect a pathological change, as it is also found in the healthy control group. Within the optic nerve, but not in the tract, there also appeared to be a direct correlation between tissue fractions and measures of disease progression. Specifically, a decrease in nWM was related to decreased visual acuity (an increase in the logMAR scale score) and a decrease in RNFL thickness.

Previous studies employing diffusion-weighted imaging to investigate white matter changes in LHON patients have demonstrated reduced FA in the optic radiation, as well as in other white matter tracts to a lesser extent [16,32,33]. Studies in the anterior optic pathway typically disclose reduced FA along at least part of the tract [15,16], although study cohorts are typically in the subacute to chronic phase of the disease. As yet, no results have been reported in the nerve itself, and diffusion parameters have been limited to those obtainable from the diffusion tensor model, based on a single diffusion weighting value.

As a final note, the utility of a more complex diffusion model is demonstrated by the fact that hypothesised correlations between diffusion variables and indicators of clinical status were observed in our acute/subacute cohort using the fractions of the three tissue compartments in the model, and not the more conventional diffusion parameters, FA and MD.

The present work suffers from a number of limitations. First, the number of LHON cases available for imaging was limited, so that statistical results should be interpreted with due caution. In particular, owing to the large number of correlations performed, results should be thought of as an exploratory analysis suggesting relationships to be tested in a larger cohort. Furthermore, LHON is a progressive disorder and the disease duration in our cohort was relatively short, so that we would not expect neurodegenerative changes to be as evident as in chronic patients. On the other hand, the fact that changes were observed hints at the sensitivity of tractography for detecting neuropathological processes.

A second limitation, common to most studies evaluating MRI in vivo, is that it was impossible to compare results to the ground truth regarding functional anatomy, defined as a specific population of neurons contributing to a specific physiological function. As stated in [29], direct interpretation of the tissue fractions described here would require comparison with histopathology. Thus, despite the advantages of the tractographic approach to neuroanatomy, its use will always be complementary to postmortem histopathology.

## 5. Conclusions

Notwithstanding the limited patient population available for study, the multishell tractography approach described was able to generate an accurate three-dimensional reconstruction of the anterior optic pathway and extract microstructural information that demonstrated differences from healthy controls and correlated as hypothesised with clinical markers of disease within the optic nerve and tract.

Using the methodology described in a larger and longitudinal study would help to elucidate the natural course of the disease, and potentially provide biomarkers of neuropathology for testing new potential treatments, such as gene therapy, for both LHON and other forms of optic atrophy [34]. The combination of multishell tractography with OCT, a complementary technique yielding quantitative insights into the natural history of LHON, may prove to be especially powerful [35].

The current work has potential implications for clinical practice as it seems to show that localised microstructural alterations, such as reduced FA values, can be detected with sufficient sensitivity to permit diagnostic inferences at the single subject level in LHON patients for whom magnetic resonance imaging findings are normal, with no apparent optic nerve or brain abnormalities. While the MRI sequence proposed is feasible for many scanner suites, the complexity of the data processing implies an important role for a facility within a tertiary-level referring centre with a high level of technical and neuroimaging competence. While the current work focused on LHON patient population, the technique is equally applicable to CNS inflammatory demyelinating diseases, such as multiple sclerosis [36]. Future work will determine whether it can accurately follow the course of the optic nerve in the presence of skull base tumours, providing crucial information in the preoperative phase of surgical treatment [37].

## Figures and Tables

**Figure 1 ijerph-19-06914-f001:**
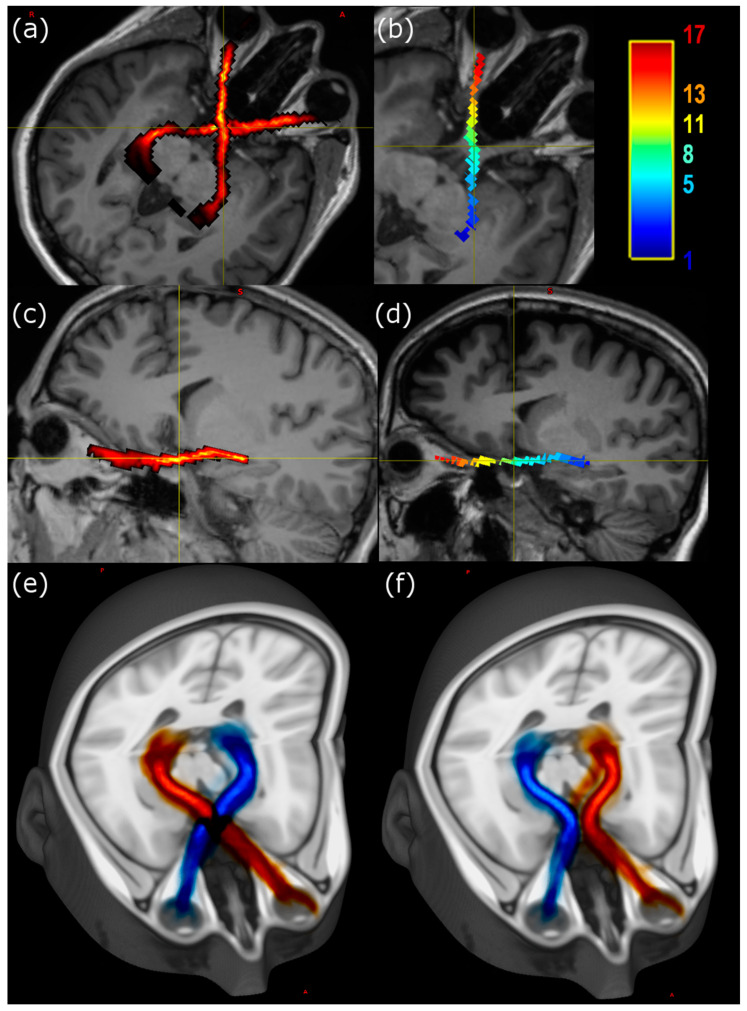
Visualisation of the anterior optic pathway. (**a**) Oblique axial slice in radiological projection showing position of tractographically determined anterior optic pathway projected onto volumetric T1-weighted image for representative participant. (**b**) Projection of segments within RL tract mapped to colour scale, as shown in scale bar. (**c**) Doubly oblique slice of the same participant aligned to the linear part of the pathway running from the left LGN. The colour scale indicates the local density of streamlines. (**d**) Projection of segments within RL tract mapped to colour scale, as shown in scale bar. (**e**,**f**) Distribution map of all participants’ crossing fibre (**e**) and noncrossing fibre (**f**) tracts projected into the standard Montreal Neurological Institute (MNI) space by linear transformation and displayed using the 2 mm T1-weighted MNI152 atlas as a background image. Fibres are displayed from left eye (blue) or right eyes (red). The width of the tracts mainly reflects anatomical variability.

**Figure 2 ijerph-19-06914-f002:**
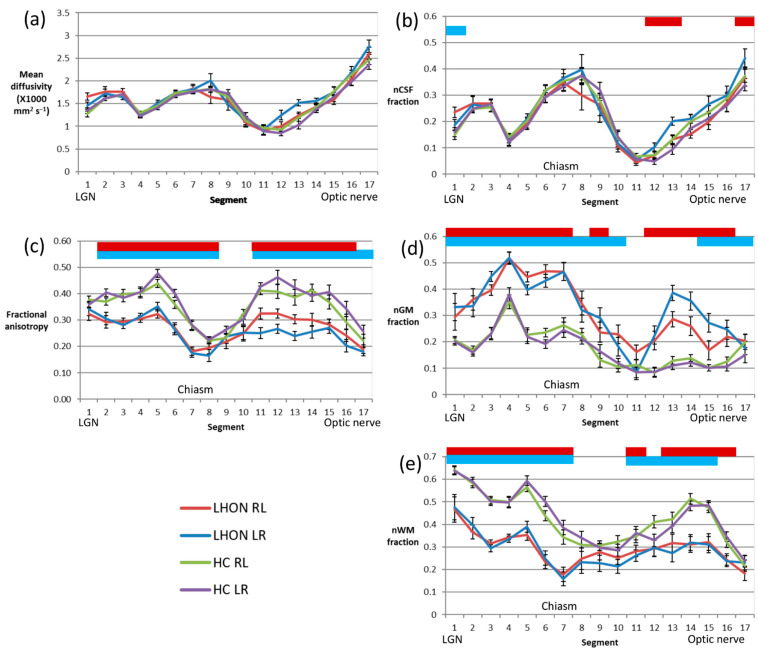
Along-tract comparison of LHON and control diffusion parameters. Summary of diffusion parameters of LHON and healthy control groups for ROIs defined by dividing each tract into evenly spaced segments within the portion between the LGN seed and chiasmatic waypoint, and in the portion anterior to the chiasma ending in the orbital terminating waypoint. (**a**) MD; (**b**) nCSF fraction; (**c**) FA; (**d**) nGM fraction; (**e**) nWM fraction. The crossing fibre tracts were evaluated for the purpose of this visualisation (LR indicates the streamlines commencing from the left LGN and terminating in the right optic disc. RL indicates streamlines that start at the right LGN and end just posterior to the left optic disc. Error bars show the standard error on the mean. Segments with a difference between LHON and healthy control (HC) groups are highlighted with a coloured bar: above red, LR tract; below blue, RL tract.

**Table 1 ijerph-19-06914-t001:** Summary of diffusion parameters of LHON and healthy control (HC) groups for ROIs defined by dividing each tract into optic nerve, and juxta- and postchiasmatic portions. All fibres in the segment closest to the chiasma were grouped as a single ROI. Mean values of patient and healthy control groups were compared using the t-test; *p*-values < 0.05 after correction for 25 multiple comparisons, highlighted in bold text, are assumed to be significant.

			LHON		HC			
Variable	ROI	Side	m	sd	m	sd	*t*	p(*t*)
**FA**	Optic tract	L	0.291	0.019	0.392	0.043	−7.49	**<0.001**
		R	0.288	0.038	0.376	0.037	−5.25	**<0.001**
	Chiasm		0.166	0.047	0.28	0.064	−4.67	**<0.001**
	Optic nerve	L	0.266	0.044	0.338	0.047	−3.57	**0.003**
		R	0.233	0.051	0.362	0.057	−5.33	**<0.001**
**MD** **(×1000 mm^2^ s^−1^)**	Optic tract	L	1.61	0.09	1.55	0.10	1.49	0.156
		R	1.64	0.21	1.57	0.13	0.84	0.422
	Chiasm		1.77	0.37	1.76	0.24	0.07	0.949
	Optic nerve	L	1.47	0.18	1.50	0.29	−0.32	0.753
		R	1.64	0.28	1.43	0.23	1.77	0.101
**nCSF ***	Optic tract	L	0.249	0.027	0.231	0.039	1.26	0.225
		R	0.244	0.060	0.238	0.054	0.22	0.828
	Chiasm		0.338	0.138	0.357	0.071	−0.36	0.726
	Optic nerve	L	0.177	0.040	0.202	0.074	−1.01	0.327
		R	0.222	0.070	0.184	0.067	1.23	0.237
**nGM ***	Optic tract	L	0.338	0.074	0.529	0.076	−5.69	**<0.001**
		R	0.324	0.108	0.514	0.076	−4.37	**0.001**
	Chiasm		0.182	0.090	0.381	0.122	−4.28	**<0.001**
	Optic nerve	L	0.272	0.092	0.372	0.060	−2.72	0.020
		R	0.256	0.096	0.357	0.061	−2.65	0.023
**nWM ***	Optic tract	L	0.407	0.066	0.233	0.075	5.56	**<0.001**
		R	0.424	0.126	0.241	0.063	3.83	0.004
	Chiasm		0.473	0.117	0.225	0.131	4.50	**<0.001**
	Optic nerve	L	0.228	0.042	0.127	0.052	4.89	**<0.001**
		R	0.268	0.068	0.121	0.046	5.43	**<0.001**

* Note: The variables nCSF, nWM and nWM represent the fraction of signal attributed to CSF-like, white matterlike, and grey matterlike signal.

## Data Availability

The data presented in this study are available on request from the corresponding author. Data used to prepare figures are available at doi: 10.5281/zenodo.6611133.

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
