# Peer review of "Multishell Diffusion MR Tractography Yields Morphological and Microstructural Information of the Anterior Optic Pathway: A Proof-of-Concept Study in Patients with Leber’s Hereditary Optic Neuropathy"

_ijerph, 2022, doi:10.3390/ijerph19116914_

Round 1
Reviewer 1 Report
I thank the invitation to review the original manuscript entitled "Multishell Diffusion MR Tractography Yields Morphological and Microstructural Information of the Anterior Optic Pathway: A Proof-of-Concept Study in Patients with Leber’s Hereditary Optic Neuropathy)" sent for publication in Int J Environ Res Public Health. The authors have evaluated in detail aspects regarding neuroimaging features about myelinated white matter tracts and nerves involving the anterior optic pathway. Tractography method is widely used in several other basic and clinical research studies and also in clinical practice in the evaluation of Hereditary Spastic Paraplegias and neurosurgical approaches. Most Leber's hereditary optic neuropathy studies have previously focused on funduscopic findings, optic coherence tomography, muscle biopsy and genetic findings, however the authors show in this original study that tractography can be also used as a potential neuroimaging biomarker during natural history, progression of visual compromise and, furthermore, as a potential marker of severity. Another important feature is that the authors have included a large number of patients with a rare disease like Leber's Hereditary Optic Neuropathy. Maybe one important point for discussion for the authors would be to provide details about the potential importance of this study in clinical practice. The manuscript is well-written, however it needs a review of language aspects by a native English speaker or a language editing and processing service.
Author Response
We thank the reviewer for his/her careful evaluation of the manuscript. We have revised the manuscript in accordance with their comments and suggestions, and below we respond to each major point:
Reviewer comment 1. "Maybe one important point for discussion for the authors would be to provide details about the potential importance of this study in clinical practice."
We agree with the reviewer that the manuscript would benefit from a discussion of the clinical relevance of the study. We have included the following text to describe the potential utility of the study for clinically-oriented research and for clinical practice (lines 470-488) and added four recent references:
Using the methodology described in a larger and longitudinal study would help to elucidate the natural course of the disease, and potentially provide biomarkers of neuropathology for the testing of new potential treatments, such as gene therapy, for both LHON and other forms of optic atrophy [34]. The combination of multi-shell tractography with OCT, a complementary technique yielding quantitative insights into the natural history of LHON, may prove to be especially powerful [35].
The current work has potential implications for clinical practice as it seems to show that localised microstructural alterations, such as reduced FA values, can be detected with sufficient sensitivity to permit diagnostic inferences at the single subject level in LHON patients for whom magnetic resonance imaging findings are normal, with no apparent optic nerve or brain abnormalities. While the MRI sequence proposed is feasible for many scanner suites, the complexity of the data processing implies an important role for a facility within a tertiary level referring centre with a high level of technical and neuroimaging competence. While the current work focused on LHON patient population, the technique is equally applicable to CNS inflammatory demyelinating diseases, such as multiple sclerosis [36]. Future work will determine whether it can accurately follow the course of the optic nerve when disturbed by the presence of skull base tumours, providing crucial information in the pre-operative phase of surgical treatment [37].
34. Carelli, V.; Karanjia, R.; La Morgia, C. Editorial: Hereditary Optic Neuropathies: A New Perspective. Frontiers in Neurology 2021, 12, doi:10.3389/fneur.2021.742484.
35. Carbonelli, M.; Morgia, C.L.; Romagnoli, M.; Amore, G.; D’Agati, P.; Valentino, M.L.; Caporali, L.; Cascavilla, M.L.; Battista, M.; Borrelli, E.; et al. Capturing the Pattern of Transition from Carrier to Affected in Leber’s Hereditary Optic Neuropathy. American Journal of Ophthalmology 2022, doi:https://doi.org/10.1016/j.ajo.2022.04.016.
36. Pisa, M.; Pansieri, J.; Yee, S.; Ruiz, J.; Leite, I.M.; Palace, J.; Comi, G.; Esiri, M.M.; Leocani, L.; DeLuca, G.C. Anterior Optic Pathway Pathology in CNS Demyelinating Diseases. Brain 2022, doi:10.1093/brain/awac030.
37. Zoli, M.; Talozzi, L.; Mitolo, M.; Lodi, R.; Mazzatenta, D.; Tonon, C. Role of Diffusion MRI Tractography in Endoscopic Endonasal Skull Base Surgery. JoVE 2021, e61724, doi:10.3791/61724.
Reviewer comment 2: "The manuscript is well-written, however it needs a review of language aspects by a native English speaker or a language editing and processing service."
The resubmitted manuscript has been completely revised for linguistic and grammatical correctness, in accordance with the journal's own Instructions for Authors, and Manuscript English Editing guidelines, by a native English speaker with many years' experience in editing scientific research reports for learned journals.
Reviewer 2 Report
In this paper the authors test whether the use of tractography MRI is helpful in visualizing the optic pathway in three dimension and can provide microstructural information demonstrating pathological variations.
Eight patients with LHON and 13 age-matched healthy controls were studied with tractography of the anterior optic pathway. The topic is interesting, the article is well written with adequate scientific method. Statistical analysis seems properly conducted.
Author Response
We thank the reviewer for his/her positive evaluation of the manuscript.
Reviewer 3 Report
This study shown that the multi-shell tractography approach was able to generate an accurate three-dimensional reconstruction of the anterior optic pathway and extract microstructural information that demonstrated differences from healthy controls and correlated as hypothesised with clinical markers of disease within the optic nerve and tract. DTI can evaluate white matter integrity in mitochondrial optic neuropathies and may yield useful surrogate biomarkers of disease severity and progression, to evaluate therapeutic efficacy in mitochondrial optic neuropathies.
Author Response

(The authors gave the same response as above.)
